# Age-Related Changes in Zinc, Copper and Selenium Levels in the Human Prostate

**DOI:** 10.3390/nu13051403

**Published:** 2021-04-21

**Authors:** Adam Daragó, Michał Klimczak, Joanna Stragierowicz, Mateusz Jobczyk, Anna Kilanowicz

**Affiliations:** 1Department of Toxicology, Medical University of Lodz, 90-151 Lodz, Poland; michal.klimczak@umed.lodz.pl (M.K.); joanna.stragierowicz@umed.lodz.pl (J.S.); anna.kilanowicz@umed.lodz.pl (A.K.); 2Department of Urology, The Hospital Ministry of the Interior and Administration, 91-425 Lodz, Poland; matijob87@gmail.com

**Keywords:** zinc, copper, selenium, prostate, human

## Abstract

Pathophysiological changes in the prostate gland—benign prostatic hyperplasia (BPH) and prostatic adenocarcinoma (PCa)—are closely related to the age of men. In the prostate gland, zinc is of particular importance for its proper functioning, especially with regard to the effects of hormonal disorders. The aim of this study was to evaluate zinc, copper and selenium concentrations in different parts of the prostate gland in relation to age and the nature of pathological changes. Zinc and copper were determined by the AAS method and selenium by the spectrofluorometric method. The concentration of zinc in the central part of the prostate increases with age, and in patients over 36 years it is twice as high as in the peripheral part, where no increase in the level of this element was observed with the age of patients. The above data confirm a possible influence of zinc on the formation of PCa (located mostly in the peripheral part of the prostate, with low levels of zinc) and BPH in the central part where the levels of this element are the highest. The results apparently confirm the disturbed homeostasis of zinc and other essential elements in the etiology of BPH and PCa.

## 1. Introduction

Two major health problems affecting aging men are benign prostatic hyperplasia (BPH) and prostate carcinoma (PCa). It should be emphasized that as proliferation of prostatic stromal cells occurs in up to 50% of men by the age of 50 and its prevalence increases with age, age is one of the most well-established factors in developing BPH [1]. PCa is one of the most commonly recognized cancers worldwide and has the second highest death rate among men, especially in developing countries [2]. While the exact cause of both BPH and PCa is unknown, diet, lifestyle and genetic background are known to be etiological factors [3,4]. These pathological processes have different histopathologies, clinical courses and metabolic alterations, and are characterized by disturbances in the homeostasis of certain trace elements, especially zinc (Zn), but also copper (Cu) and selenium (Se) [5].

The prostate gland consists of three glandular zones, namely a central (CZ), peripheral (PZ) and transition zone (TZ), with widely differing susceptibilities to PCa and BPH. PZ is the largest part of the prostate, constituting 70% of the gland; it is responsible for the production and secretion of Zn and citrate and is the most frequent site of PCa. In contrast, TZ (which accounts for 2–5% of the gland) is the almost exclusive site of BPH [6]. The normal human prostate accumulates the highest levels of Zn of any soft tissue in the body, because Zn is an essential trace element required for proper prostatic gland function. Its presence inhibits mitochondrial aconitase activity, i.e., limiting the oxidation of citrate and mitochondrial terminal oxidation and respiration, and has anti-proliferative effects, such as induction of mitochondrial apoptogenesis and suppression of NFκB activity [7,8,9]. Interestingly, while many studies indicate that Zn levels are significantly disturbed in such states as BPH or PCa, the role of Zn in the etiology of these proliferative changes is not fully known. Many studies indicate that Zn concentration in malignant prostate tissues can be as much as 85% lower than in unchanged prostate tissue [5,10,11,12]. However, numerous studies have described a significant, even several-fold increase of Zn levels in BPH tissues. Unfortunately, serum Zn concentration does not seem to be a reliable biomarker of Zn status in the body, and does not reflect its level in the prostate. Some data indicate a decreased Zn concentration in PCa patients [13,14,15,16,17], while other data suggest increased Zn concentration [18,19] or no association at all [20]. Therefore, post-mortem analysis of prostate tissue seems to provide a more accurate assessment of the role of Zn in the development of both PCa and BPH.

Cu and Se also play an important role in the development of prostate hyperplasia. Disturbances in Cu homeostasis are evidenced by increased serum concentrations in various malignancies [21,22]. Elevated Cu levels have also been reported in both BPH and PCa samples [12]. This effect may be related to the involvement of Cu in cellular proliferation, i.e., activation of angiogenic growth factors [23].

A key role of Se in the proper functioning of the prostate is believed to be associated with its protective effects, particularly its antioxidant and antiaging activities, and anti-tumor potential. It has also been associated with many degenerative conditions, including inflammation and the carcinogenic process in the prostate tissue [24]. The role of Se in the prostate is confirmed by our own previous studies, which found its level to be about 60% higher in both BPH and PCa compared to control tissue [12].

The aim of this study was to examine the age dependency of changes in Zn concentration in different parts of the prostate. It also examines the concentration of Cu, due to the close relationship between Zn homeostasis and Cu homeostasis, and of Se, because of its use in anticancer supplementation. It also compares the findings with those from several samples of prostates from men with BPH and PCa.

## 2. Materials and Methods

The study was conducted on human prostate tissue obtained from autopsy specimens of 147 patients aged 20 to 83 years. From each patient, a cross section of the prostate was taken. Before the analysis, all tissues underwent pathomorphological evaluation to exclude or confirm proliferative changes. The obtained prostates were divided into central (periurethral) and peripheral parts. All samples were stored frozen (−80 °C) in polypropylene anti-contamination vessels before the elements were analyzed.

Table 1 shows the number and age structure of the investigated groups. The samples were divided into three groups depending on the histopathological changes of the investigated samples: a control group (no histopathological changes), benign prostate hypertrophy group (BPH) and adenocarcinoma of the prostate group (PCa). The control group was additionally divided according to age into control group 1 (≤35 years) and control group 2 (≥36), as 35 years is considered the beginning of andropause in men [25].

Zn and Cu levels were determined by flame atomic absorption spectrometry (GBC Avanta PM) following mineralization. The limits of detection (LODs) were 0.024 µg/mL for Zn and 0.0678 µg/mL for Cu; these were calculated as concentrations corresponding with an absorption value equal to a three-fold standard deviation of the signal for the lowest concentration of standard. A calibration curve was prepared using different concentrations of various metal solution (ASTASOL-Mix, ANALYTIKA^®^, spol. s r.o., Praha, Czech Republic). The obtained curves were linear in the following concentration range: 0.05–1.6 µg/mL for Zn and 0.05–5 µg/mL for Cu. Their equations were:y = 0.2048x + 0.0051; R^2^ = 0.9970 (Zn)
y = 0.0789x + 0.0005; R^2^ = 0.9981 (Cu).

Se determinations in samples were performed on a spectrofluorometer (Hitachi F-4500) according to the method described by Danch and Drozdz (1996) [26]. LOD was 0.013 µg/mL. The obtained curve was linear in the concentration range of 0.05−1.6 µg/mL, and its equation was:y = 212.3211x + 6.95214; R^2^ = 0.9984 (Se).

Intralaboratory quality control of the determination was based on the certified standard of freeze-dried bovine liver SRM 1577b (National Institute of Standards and Technology, Gaithersburg, MD, USA), which contained certified concentrations of the elements (µg/g wet tissue): Zn (127 ± 16), Cu (160 ± 8) and Se (0.73 ± 0.06). The mean discrepancies between the results obtained and the certified values expressed as relative standard deviation (RSD) were as follows: Zn ± 0.25%, Cu ± 6.8% and Se ± 3.8%.

Concentrations of Zn, Cu and Se in investigated tissues were expressed as µg/g wet tissue. STATISTICA 13.0 (StatSoft Inc., Tulsa, OK, USA) was used for all statistical analyses. After performing a one-way ANOVA, the significance of the differences for the selected parameters was set using Tukey’s test, following Bartlett’s test for homogeneity of variance. Differences with a *p*-value of less than 0.05 were considered statistically significant. Pearson correlation coefficient test was applied to assess univariate associations.

The study was approved by the Ethics Committee for Scientific Research at the Medical University of Lodz, Poland (RNN/39/16/KE).

## 3. Results

Table 2 presents concentrations of analyzed elements in the central and peripheral parts of the prostate in the selected groups. Only the group of younger men (≤35 years) demonstrated similar levels of Zn in both parts of the prostate (84.59 ± 23.68 µg/g wet tissue in the central part and 114.44 ± 37.97 µg/g wet tissue in the peripheral part) and did not differ significantly. In persons over 36 years of age, the level of Zn in the central part of the prostate was three times higher than the younger group, with no changes observed in the peripheral part. In addition, in this group, the concentration of Zn was more than twice as high in the central part than the peripheral part. Additionally, to highlight the nature of the changes occurring with age, a strong correlation (r = 0.8) was observed between the ratio of the Zn concentrations in the central and peripheral parts and the age of the people from whom the samples were taken (Figure 1). These results indicate that age has a close relationship with change of Zn ratio in the prostate: this ratio increases with age.

The highest level of Zn was observed in the central part of the prostate in the group with histopathologically confirmed BPH; this value was significantly higher than in the other groups. A slight decrease in Zn levels was observed in the peripheral part of the prostate in this group, although this value was not different from both control groups. The opposite effect was observed in samples with histopathologically confirmed PCa: while the level of Zn in the central part of the prostate did not differ from those of the healthy controls over 35 years of age, it dropped dramatically to below 30 µg/g wet tissue in the peripheral part of the prostate. In addition, Zn ratios between the central and peripheral parts were over four and eight for the BPH and PCa groups, respectively. These findings clearly indicate the great degree of variation in the Zn profile with regard to histopathological changes.

No statistically significant differences in Cu levels were found with regard to age in the control samples (Table 2). In the case of samples taken from individuals with histopathological changes in the prostate, a statistically significant increase in Cu levels of about 30% was observed in the central part, while in the peripheral part, its concentration decreased by 30% in the BPH group and more than 50% in the PCa group. As in the case of Zn, the ratio of Cu concentrations in both examined parts of the prostate seems to be a very good indicator of presented differences. This ratio was increased by 100% in the case of samples with BPH and 200% in persons with PCa in relation to controls.

Se concentrations in the BPH group were significantly different to other groups (Table 2). In the central part, the Se concentration was 40% higher than in tissues without proliferative changes, while in the peripheral part, it was almost 30% lower. As with the other elements, this resulted in a statistically significant increase in the ratio of this element in both examined parts of the prostate.

## 4. Discussion

It has been well documented that the prostate accumulates high amounts of Zn [3,27] and that this element is not uniformly distributed in the prostate gland [28]. This non-uniform distribution was first reported by Gyorkey et al. (1967), who identified the highest concentration in the lateral zone of human prostate, followed by the dorsal zone, i.e., the PZ according to the current anatomical classification, and lower concentrations in the inner and anterior zones [29]. Several decades later, Leitao et al. (2009) reported that the PZ contains more Zn than the TZ, but they did not provide specific levels [30]. Both studies used prostate from men under 36 years, without any hyperplasic changes, as the controls. The results of this study indicate that in the younger age group (≤35 years), both the central and peripheral parts demonstrate nearly comparable Zn concentrations, with the levels being a little higher, although not significantly so, in the peripheral part. However, in individuals aged over 36 years, the central part accumulates over twice the amount of Zn than the peripheral part. Our present findings also indicate that Zn content increases with age in the central part, but not in the peripheral part (Figure 2 and Figure 3).

A similar tendency, but throughout the whole prostate, has also been observed in other studies, as summarized by Zaichick and Zaichick (2014) [31]. The Zn content in the prostate of men aged over 50 years is around 1.5–3.0 times higher than in those aged under 30 years. They also propose that this age-related increase in prostatic Zn content might be partially connected with the accumulation of prostatic fluid observed in men aged between 30 to 50 years, as prostatic fluid contains high amounts of Zn [31].

The differences of Zn concentration between zones or parts became more remarkable when analyzing samples from individuals suffering from hyperplasic changes of the prostate, i.e., BPH and PCa. In the case of PCa, the concentration of Zn in peripheral part was around three times lower than in controls. On the other hand, Zn concentrations in the central part were comparable to that of controls aged over 40 years. The reduction in Zn in the PCa tissues, compared to normal and benign prostate tissue, is a well-known phenomenon and has been well documented by many researchers [12,27,32,33]. Our findings indicate that this phenomenon stems mainly from a decrease in Zn in the peripheral part of the gland, i.e., where the malignancy of the prostate gland develops [27].

Costello and Franklin (1998) propose that the mechanism of prostate malignancy may be associated with changes in the Zn-citrate metabolism. Zn is an inhibitor of mitochondrial aconitase in the prostate, so malignant prostatic cells, which have lost the ability to accumulate Zn, switch from citrate-producing cells into citrate-oxidizing ones, thus satisfying the increased energy demands for cancer cells [28]. However, Zn plays also a role in an apoptosis. It induces the expression of Bax with its incorporation into the membrane of mitochondrion, which in turn induces the release of cytochrome c and caspase-promoted apoptosis. Therefore, the reduction in Zn levels observed in PCa results also in decreased apoptosis of the malignant cells [27]. Downregulation of the ZIP (Zrt-Irt-like protein) family transporters, these being Zn uptake transporters, results in Zn depletion in malignant cells [27]. ZIP mediates the transport of Zn from the extracellular fluid and of intracellular vesicles into the cytoplasm, and several studies have reported significant downregulation of ZIP1, ZIP2, ZIP3 and ZIP4 in PCa compared to normal prostate [3].

While a fall in Zn level has been confirmed by dozens of studies in the case of PCa, the results are more ambiguous in the case of BPH; studies have found the Zn level to be increased [12], decreased [14] and even unchanged in BPH [34] compared to unchanged prostatic tissue. In the present study, the Zn concentration in BPH was the highest of all studied groups in the central part of the hyperplasic tissue, and similar to both control groups in the peripheral part. Zn is a known inhibitor of 5α-reductase, converting testosterone into dihydrotestosterone (DHT) and DHT is one of the main factors contributing to the development of BPH [35,36]. Therefore, the presence of increased Zn content in the central part may be some kind of “protective mechanism” from cell proliferation induced by DHT. However, the source of Zn in the central part remains a mystery. It is likely that the Zn is obtained from a general pool rather than the other parts of the prostate, as the concentration of Zn in the peripheral parts did not appear to be significantly diminished.

In the case of Cu, no age-related differences were observed in prostatic Cu content in men aged 21–87 years [37], nor with regard to the PZ of unchanged prostates [38]. Our present findings indicate that while the concentrations of Cu in both zones are comparable and are not age-dependent in controls, changes in Cu level were present in the BPH and PCa groups. In both groups, Cu content was decreased in the peripheral part, but increased in the central part, suggesting a disturbed homeostasis in these two conditions.

Some studies have reported an accumulation of Cu in PCa samples [21,39], while Sapota et al. (2009) reported increased Cu content in PCa, with the highest values in the BPH group [12]. The latter findings seem to be in line with those of the present study, which indicate higher Cu concentration in the BPH group than in the PCa group for both parts of the prostate. In contrast, Jain et al. (1994), Denoyer et al. (2015), and Zaichick and Zaichick (2016) report a higher Cu content in PCa tissues than control and BPH tissues [21,39,40]. This relationship seems to be a complex one, as Denoyer et al. (2015) noticed a considerable range of Cu concentration in human PCa samples, with half of the samples having a greater Cu concentration than the BHP samples [40]. There is clearly still a need for further studies regarding the level of Cu content in the prostate, particularly because one emerging area of PCa therapy is based on the use of Cu-ionophores to elevate intracellular bioavailable Cu in malignant cells [41,42].

The last element analyzed in this study was Se, whose supplementation is often suggested because of its anticancer properties [43,44]. Our findings suggest that Se content in the different parts of the prostate was not age-dependent, and no differences were found between the two analyzed parts in unchanged prostates. In healthy prostates, the Se concentrations from men aged <41, 41–60 and >60 years were found to be comparable [37]; however, Se content was found to be greater in prostatic tissue from BPH groups than normal prostates [34,45,46]. Only a subtle increase of Se level was observed in the present study, and only in the central part, with a concomitant decrease in the peripheral one. A similar pattern was observed in the case of PCa, but the Se concentration in the central part was within the range of the control samples. Nyman et al. (2004) also observed differences in Se content between prostate zones: the PZ of PCa samples accumulated more Se than the TZ [47]. Other studies on the Se content in PCa samples are more contradictory: Zaichick and Zaichick (2012) report the lower concentrations of Se in PCa compared to BPH [45], while Zachara et al. (2005) report a higher level compared to BPH and normal prostates [48]. It seems that the role of Se in proliferative changes of the prostate is a more complex one than first thought.

We are aware that our study has some limitations, such as the lack of complete patient data due to the way of obtaining the prostate samples (post-mortem) or the lack of analysis of biochemical parameters related to elemental metabolism in tissues.

There is therefore undeniable evidence that Zn concentrations differ between the studied parts of the prostate. Moreover, our study is the first to report that Zn concentration in the central part increases with age. Zn concentration in this part of the prostate from patients with BPH is even higher. On the other hand, Zn concentration in the peripheral part of the prostate does not change with age and the lowest concentrations were noted in PCa samples. Zn ratios between the central and peripheral parts seem to be a good indicator reflecting described changes in the prostate in relation with both age and histopathological changes of this gland. In addition, our findings indicate no age-related changes in Cu and Se.

## Figures and Tables

**Figure 1 nutrients-13-01403-f001:**
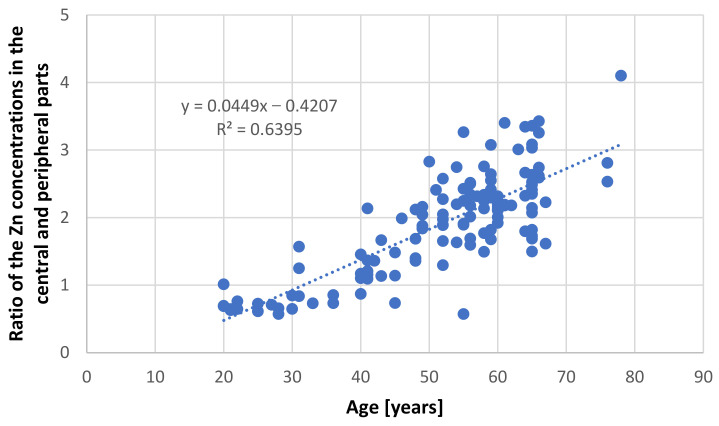
Relationship between ratio of zinc concentration in the central and peripheral parts of the prostate and age in samples taken from persons without proliferative changes.

**Figure 2 nutrients-13-01403-f002:**
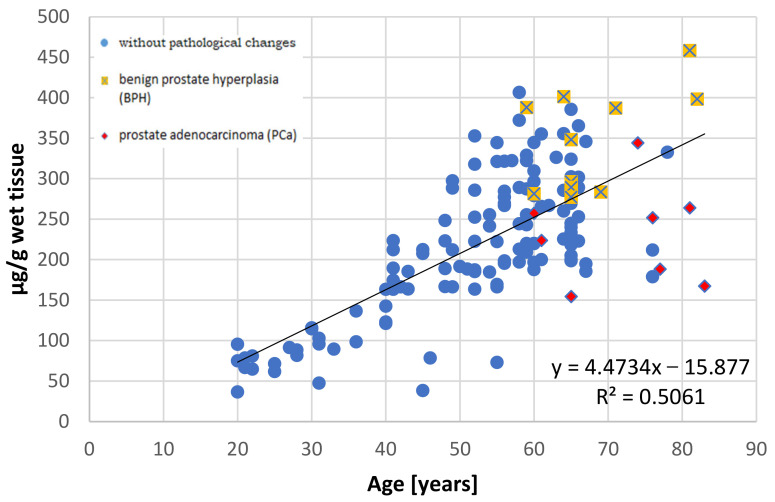
The relationship between zinc concentration (µg/g wet tissue) and age (years) in the central part of the human prostate with regard to histopathological status.

**Figure 3 nutrients-13-01403-f003:**
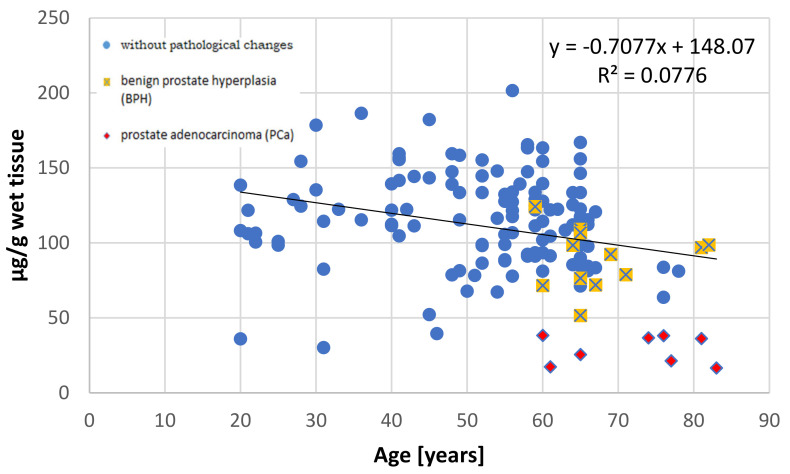
The relationship between zinc concentration (µg/g wet tissue) and age (years) in the peripheral part of the human prostate with regard to histopathological status.

**Table 1 nutrients-13-01403-t001:** Characteristics of the investigated groups.

Group	N	Age	
		Mean ± SD [years]	Range [years]
Control Group 1 (age ≤ 35)	20	26.8 ± 5.2	20–35
Control Group 2 (age ≥ 36)	108	56.6 ± 8.5	40–78
BPH	11	68.0 ± 7.5	59–82
PCa	8	72.1 ± 9.0	60–83

**Table 2 nutrients-13-01403-t002:** Zinc, copper and selenium concentrations (µg/g wet tissue) in the central and peripheral parts of the prostate in age groups with no proliferative changes and in the groups with BPH and PCa. All values are expressed as means ± SD.

	Control Group 1 (Age ≤ 35)	Control Group 2 (Age ≥ 36)	BPH	PCa
**N**	20	108	11	8
**Zn**				
**Central Part**	84.59 ± 23.68	240.01 ± 69.02 ^a^	350.59 ± 61.64 ^ab^	231.43 ± 61.88 ^ac^
**Peripheral Part**	114.44 ± 37.97	117.20 ± 29.97 ^d^	87.95 ± 20.13 ^d^	28.76 ± 9.57 ^abcd^
**Central/Peripheral Ratio**	0.79 ± 0.24	2.12 ± 0.63 ^a^	4.10 ± 0.78 ^ab^	8.49 ± 2.30 ^abc^
**Cu**				
**Central Part**	1.08 ± 0.14	1.19 ± 0.14	1.52 ± 0.08 ^ab^	1.40 ± 0.08 ^abc^
**Peripheral Part**	1.02 ± 0.14	1.03 ± 0.11	0.72 ± 0.08 ^abd^	0.45 ± 0.09 ^abcd^
**Central/Peripheral Ratio**	1.07 ± 0.07	1.16 ± 0.14	2.12 ± 0.23 ^ab^	3.21 ± 0.76 ^abc^
**Se**				
**Central Part**	0.18 ± 0.05	0.19 ± 0.06	0.27 ± 0.04 ^a^	0.23 ± 0.02
**Peripheral Part**	0.15 ± 0.04	0.13 ± 0.06	0.10 ± 0.02 ^d^	0.16 ± 0.02 ^cd^
**Central/Peripheral Ratio**	1.15 ± 0.18	1.56 ± 0.40	2.68 ± 0.64 ^ab^	1.46 ± 0.26 ^c^

Statistically significant differences at *p* ≤ 0.05. a—Results statistically significant compared to ≤35 group, b—Results statistically significant compared to ≥36 group, c—Results statistically significant compared to BPH group, d—Results statistically significant compared to the central part.

## Data Availability

The data presented in this study are available on request from the corresponding author. The data are not publicly available due to privacy and ethical restrictions.

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
