# Peer review of "Age-Related Changes in Zinc, Copper and Selenium Levels in the Human Prostate"

_nutrients, 2021, doi:10.3390/nu13051403_

Round 1
Reviewer 1 Report
The study was designed to investigate the concentration of minerals in different zones of prostate tissue and their correlation with age. The study was very interesting and could provide informative implications on how diet affects the development of pathological condition in prostate as well as new perspectives in terms of prostate cancer prevention.
There are a few minor points that authors should address:
- Statistical analysis: Ln 109: should be “one way ANOVA” instead of “ANOVA”
- Ln 110: What was the rationale that Bartlett’s test for homogeneity of variance was used? Did the data come from a normal distribution?
- Table 3, statistical analysis should be provided for the ratio of minerals
- The authors should provide the limitations of the study in the discussion session.
- The authors should also discuss how to interpret the differences of their results compared to other published studies. Apparently, not all published papers showed the same trend in terms of mineral levels in prostate tissues.
Author Response
Dear reviewer,
thank you very much for your insightful evaluation of our work and submitting your comments. We hope that the answers below will be satisfactory to you and allow you to evaluate our work positively.
- Thank you for this comment, it was corrected (LN: 109).
- Bartlett's test is well suited to compare groups with a normal distribution, which differ significantly in size, but the lowest number of samples of all groups have to be ≥5. All these premises were satisfied in our study.
- Thank you for this comment, but according to the comments of other reviewer we decided to remove this table and its description in text.
- In the Discussion section, we added the limitations of the study (LN: 255-257).
- In the discussion section we tried to discuss existing data on tested elements in prostate tissues. While it is possible in the case of histopathologically altered tissues, there are no data in the available literature on the levels of studied elements in the different parts of the prostate of healthy people. The differences in the results obtained by other researchers and our results are difficult to discuss. These changes could stem from different methods of obtaining prostate samples (for example TURP), preparation of the samples and analytical techniques of elements determination (AAS, X-ray etc.). The main aim of this study was to check the levels of the studied elements in pathologically unchanged tissues from men of different age. This is the first time this study was done and it has no reference in literature.
Reviewer 2 Report
In this work, age related alterations of zinc, copper and selenium levels in a cohort of samples including normal, BPH and prostate cancer. There is no mechanistic analysis or experiments. In my review, this work can be accepted in its current format as a report or Short communication.
A proofreading by a native speaker is needed.
Author Response
Dear Reviewer,
thank you kindly for reviewing our work and posting your comments. However, we could not agree with you to published our paper as short communication. Please note that despite time consuming collection of samples we obtained more than 100 samples and presented data are unique in terms of providing concentration of elements in various parts of prostate. Thank you for the suggestion of proof-reading. The paper was checked by a native speaker and we send a certificate of proof-reading to the Editor.
Reviewer 3 Report
The manuscript by Darago et al. provides an analysis of the changes in the levels of zinc, copper, and selenium in the central and peripheral human prostate as a function of age and histology. The analyses are generally carefully performed and well described. Although the data provided is descriptive, the approach of studying both prostate zones and the interest in the selected elements makes this an interesting paper. This is particularly true for zinc, known to be a critical regulator of prostate physiology and whose levels are altered during progression to malignancy of that organ.
Minor issues:
- The authors present relative levels of the chosen elements as fractions (Table 3) which can be confusing. They might consider more conventional means of representing the relative levels and provide a better description of how the ratios were generated.
- The rationale for looking at the ratios of the studies elements was never provided. This is significant as perhaps conclusions can be suggested regarding shared or distinct changes in transport of these elements into the prostate.
- Statistical significance was indicated as being a p-value of less that 0.05. Actual statistical values, or at least ranges should be provided, especially given the relatively small sample sizes.
Author Response
Dear reviewer,
thank you very much for evaluating our work and for the comments you have posted. We hope that the responses below will be sufficient for you and allow us to publish our paper.
1 and 2. Thank you very much for these comments. We looked once more at these results with a critical eye and we get to the conclusion that interpretation of ratios really could be confusing and we don’t want the reader to get to any misleading conclusions because of that. Therefore we decided to remove these results (table 3 and its description in the text). The main changes of analyzed elements and their course are better visible from the table
3. Thank you for your suggestion, but we decided to not provide actual statistical values, because it could made the table 2 much bigger (too much digits near means) and therefore hard to analyze and a little bit confusing. We are aware that the groups differ significantly in numbers and we have small groups, but we use Barlett’s test, which is appropriate in this kind of situation. Additionally, we added once more under the table 2, at which p we evaluate our results as statistically significant.